# Experiences of Black and Latinx health care workers in support roles during the COVID-19 pandemic: A qualitative study

Zorimar Rivera-Núñez[1]*, Manuel E. Jimenez[2,3], Benjamin F. Crabtree[3], Diane Hill[4], Maria B. Pellerano[3], Donita Devance[5], Myneka Macenat[3], Daniel Lima[2], Marsha Gordon[3], Brittany Sullivan[3], Robert J. Rosati[6], Jeanne M. Ferrante[3], Emily S. Barrett[1], Martin J. Blaser[7], Reynold A. Panettieri, Jr.[8], Shawna V. Hudson[3]

1 Department of Biostatistics and Epidemiology, Rutgers School of Public Health, Environmental and Occupational Health Sciences Institute, Rutgers University, Piscataway, New Jersey, United States of America, 2 Department of Pediatrics, Rutgers Robert Wood Johnson Medical School, New Brunswick, New Jersey, United States of America, 3 Department of Family Medicine and Community Health, Rutgers Robert Wood Johnson Medical School, New Brunswick, New Jersey, United States of America, 4 University-Community Partnerships, Rutgers School of Public Affairs and Administration, Rutgers University, Newark, New Jersey, United States of America, 5 University-Community Partnerships, Rutgers University, Newark, New Jersey, United States of America, 6 Connected Health Institute, VNA Health Group, Holmdel, New Jersey, United States of America, 7 Departments of Medicine and Pathology, Rutgers Robert Wood Johnson Medical School, Rutgers Center for Advance Biotechnology and Medicine, New Brunswick, New Jersey, United States of America, 8 Rutgers Institute for Translational Medicine and Science, Rutgers Robert Wood Johnson Medical School, New Brunswick, New Jersey, United States of America

* zorimar.nunez@rutgers.edu

## Abstract

Black and Latinx individuals, and in particular women, comprise an essential health care workforce often serving in support roles such as nursing assistants and dietary service staff. Compared to physicians and nurses, they are underpaid and potentially undervalued, yet play a critical role in health systems. This study examined the impact of the coronavirus disease 2019 (COVID-19) pandemic from the perspective of Black and Latinx health care workers in support roles (referred to here as HCWs). From December 2020 to February 2021, we conducted 2 group interviews (n = 9, 1 group in English and 1 group in Spanish language) and 8 individual interviews (1 in Spanish and 7 in English) with HCWs. Participants were members of a high-risk workforce as well as of communities that suffered disproportionately during the pandemic. Overall, they described disruptive changes in responsibilities and roles at work. These disruptions were intensified by the constant fear of contracting COVID-19 themselves and infecting their family members. HCWs with direct patient care responsibilities reported reduced opportunities for personal connection with patients. Perspectives on vaccines reportedly changed over time, and were influenced by peers' vaccination and information from trusted sources. The pandemic has exposed the stress endured by an essential workforce that plays a critical role in healthcare. As such, healthcare systems need to dedicate resources to improve the work conditions for this marginalized workforce including offering resources that support resilience. Overall working conditions and, wages must be largely improved to ensure their wellbeing and retain them in their roles to manage the next public health emergency. The role of HCWs serving as ambassadors to

**Data Availability Statement:** All relevant data are within the paper and its Supporting information files.

**Funding:** This work was supported by the National Center for Advancing Translational Sciences UL1TR003017 at the National Institutes of Health. The funders had no role in study design, data collection and analysis, decision to publish, or preparation of the manuscript.

**Competing interests:** I have read the journal's policy and the authors of this manuscript have the following competing interests: Research reported in this publication was supported by grant UL1TR003017 from the National Center for Advancing Translational Sciences at the National Institutes of Health (NIH). NIH had no role in the design and conduct of the study; collection, management, analysis and interpretation of the data; preparation, review or approval of the manuscript; and decision to submit the manuscript for publication. Ms. Pellerano reports grant funding from Johnson & Johnson Corporate Foundation, personal fees from the Patient Centered Outcome Research Institute for grant reviewing, and personal fees from the University of Massachusetts, Lowell for participation on a grant advisory committee. These sources provided salary support for Ms. Perellano but did not have any additional role in this study design, data collection and analysis, decision to publish, or preparation of the manuscript. This does not alter our adherence to PLOS ONE policies on sharing data and materials. The authors have no other financial interests to disclose.

provide accurate information on COVID-19 and vaccination among their coworkers and communities also warrants further study.

## Introduction

As of June 2021, the United States continues to have the highest number of coronavirus disease 2019 (COVID-19) cases and deaths in the world. Individuals working in healthcare settings, the frontline of the pandemic, face higher risk of infection as well as higher risk of psychological stress compared to the general population [1–4]. While nurses and physicians are the most recognized frontline workers, there are a variety of other roles in healthcare including certified nurse assistants, therapists, emergency medical service personnel, dietary and food services staff, and administrative staff, among others, referred to here as health care workers (HCW) [5]. HCWs work alongside physicians and nurses but are less recognized and lower paid. This essential workforce comprises nearly 7 million people in low-paid jobs including healthcare support workers (e.g., dietary aids), direct care workers (e.g., certified nursing assistants), and healthcare service workers (e.g., hospital janitor staff) [6]. Furthermore, in the U.S., the vast majority of support HCWs are women (>80%) and they are disproportionately Black and Latinx [7, 8].

In many cases, these HCWs have different relationships with their communities than other health care professionals. They often live in the same communities that they serve and have established ties with community members. Many times, they share ethnicity, language, socio-economic status and life experiences with the community members they serve. These HCWs have played an immense role in the COVID-19 response. For example, in collaboration with community-based organizations, HCW have contacted socially isolated patients, connecting them with sources of critical important care and support [9]. They also served in hospital and nursing homes settings such as laboratory technicians, phlebotomists or therapists in direct contact with COVID-19 patients. Their role is even more important in underserved and minority populations where these workers break barriers of culture, language, and isolated neighborhoods and community hospitals to deliver health care and social and public health interventions.

Overall, individuals from racial and ethnic minority populations have been disproportionately impacted by the pandemic. For example, the risk of hospitalizations and death from COVID-19 are 2–3 times higher among Black and Latinx compared to White individuals [10–12]. Structural and institutional racism underlies their higher risk and impact their ability to avoid infection and seek care [13]. Black and Latinx HCWs are not exempt from the profound effects of these factors. Furthermore, the interaction between these factors and the inherent higher risk of COVID-19 due to their profession, make Black and Latinx HCWs a population that is at particularly acute risk. To date only a few studies have examined the perspectives of HCWs during the COVID-19 pandemic [2, 14, 15], and none have focused on their dual perspective as high-risk workers and members of marginalized communities, which has the potential to yield critical insights for equity promotion during this pandemic and future public health crises. This study examined the experiences of Black and Latinx HCWs to understand how the pandemic impacted their profession, job responsibilities, and relationship with their communities.

## Methods

As part of NJ HEROES TOO (**N**ew **J**ersey **H**ealthcare **E**ssential Worker **O**ut**R**each and **E**ducation **S**tudy- **T**esting **O**verlooked **O**ccupations), we conducted group- and one-on-one

interviews online [16]. This study was part of the NIH Rapid Acceleration of Diagnostics Underserved Populations (RADx-UP) Initiative which aims to understand disparities in underserved populations, with particular focus on COVID-19 testing [17]. We purposively sampled Black and Latinx individuals who worked as staff for 4 health care employers in New Jersey, including both in-patient (2 urban university hospitals) and outpatient (long-term care and homecare) settings in 4 counties with high numbers of Black/Latinx populations and COVID-19 burden. Employees over age 18 years who identified as Black or Latinx and identified English or Spanish as their primary language were eligible.

## Data collection

We recruited 23 HCW, 3 were not eligible and 3 did not attend. We conducted 2 group- and 8 individual interviews with Black and Latinx HCW (N = 17) between December 2020 and February 2021 using a secure Zoom platform. Variation in HCW work schedules made group interviews largely prohibitive. After completing 2 group interviews, we began recruiting participants for individual interviews in order to accommodate their schedules and ensure our approach was responsive to their needs. Group interviews were led by a primary and secondary facilitator and included two study team members for notetaking and technical assistance. We used a semi-structured interview guide for group- and individual interviews, which the team iteratively developed through literature review, prior experience, and debrief meetings after initial interviews. All interviews were recorded and transcribed verbatim. Group interviews lasted approximately 90 minutes, and individual interviews 20–30 minutes. In conducting the interviews over time and interview format, perspectives expressed by the respondents were remarkably consistent, thus leading to our conclusion of reaching thematic saturation.

## Data analysis

We analyzed the interview data using an "editing" approach as it was collected [18]. The study team debriefed after each interview and met weekly to review the data and discuss emerging themes. We initially read each transcript openly, and then in a second reading, we cut and pasted meaningful segments of text into approximately 4-page summaries. These summaries highlighted themes that emerged specifically from each interview, independent of the other interviews. We then comparatively analyzed the summaries to identify cross-cutting themes.

This study was approved by the Rutgers Biomedical Health Sciences IRB and follows the Standards for Reporting Qualitative Research [19]. All participants provided verbal consent prior to participation.

## Results

Table 1 summarizes demographics and job descriptions of our sample. The median age was 48 years (range 25–58); 47% were Black, 53% Latinx, and 100% female.

We identified three key themes that provide insight into the dual experiences of HCWs as high-risk workers and members of marginalized communities during the COVID-19 pandemic: (1) Profound impact of the pandemic on job responsibilities, work settings, and personal connections, (2) Fear and uncertainties caused by the pandemic; and, (3) Shifts in testing frequency and vaccine attitudes as the pandemic evolved.

### Profound impact of the pandemic in job responsibilities, work settings, and connections

HCWs described the substantial impacts of the pandemic on their personal lives.

**Table 1. Demographic characteristics of study participants.**

|  | Total (n = 17) | |
|---|---|---|
|  | **n** | **Percent** |
| **Age** | | |
| Median | 48 | |
| Range | 25–58 | |
| **Sex** | | |
| Female | 17 | 100.0 |
| **Race/Ethnicity** | | |
| Black or African American | 8 | 47.1 |
| Hispanic/Latino-White | 0 | 0 |
| Hispanic/Latino-Other | 9 | 52.9 |
| **Education** | | |
| Doctoral/Professional Degree | 1 | 5.9 |
| Master's Degree | 2 | 11.8 |
| 4-Year Degree | 6 | 35.3 |
| Associate Degree | 1 | 5.9 |
| Some College, No Degree | 3 | 17.6 |
| High School Diploma/Equivalent | 3 | 17.6 |
| < High School | 1 | 5.9 |
| **Household Income** | | |
| <$120,000–100,000 | 2 | 11.8 |
| $75,000 to 99,999 | 1 | 5.9 |
| $50,000 to 74,999 | 6 | 35.3 |
| $25,000 to 49,999 | 5 | 29.4 |
| < $25,000 | 1 | 5.9 |
| Refused/Missing | 2 | 11.8 |
| **Total Household Members** | | |
| 1 | 4 | 23.5 |
| 2 | 0 | 0.0 |
| 3 | 5 | 29.4 |
| 4 | 4 | 23.5 |
| 5 | 2 | 11.8 |
| 6+ | 2 | 11.8 |

"Well, the impact has been tremendous for me. . .. I lost my husband during COVID, and I think that—in May, that was kind of the height of the COVID, so I couldn't do a lot of things. And I still had to work and so forth and take care of the family and try to make them safe."

[Group 1, community health worker in home setting]

They described the distress caused by knowing they had contact with positive cases.

"By the middle of April, I'd lost four friends to it. . . we were altogether Friday, Saturday, Sunday, and they all caught it, and I was spared. I mean, I was spared"

(Participant 8, outpatient care staff, hospital setting)

Experiences in the workplace intensified this impact. Participants shared stories about the changes they endured at their place of employment. These changes varied based on work setting and job roles and responsibilities. Participants in positions without direct patient contact who were physically at their place of employment described abrupt structural changes in regular job arrangements and duties to comply with the stricter COVID-19 preventive measures.

"[O]ur department was actually closed March 20th, and we were put into what they call. . . a labor pool. So our jobs ranged from either mask distribution, cleaning COVID vents, temperature taking. . . So a variety of different roles that we played outside of what our normal job duties would be."

[Participant 9, therapy department staff, hospital setting]

HCWs reported that they needed to adapt quickly, not only to aggressive prevention measures, but also to rapidly changing work-related tasks and expectations.

"As far as work goes, it's been implementing new policies every day to the workflow being changed drastically".

[Participant 4, administrative support staff, hospital setting]

They commented on experiences from friends and community members about the long hours and lack of personal protective equipment.

"I have a friend of mine that work at the hospital. She works 16 hours. Sixteen hours. She said [name of participant], I barely have no time to eat because she's standing on her feet to try to help the best way she can. Sometimes they have no equipment in there. They have no gloves, no masks. She said [name of participant], I have to look all over the place to find equipment so I could help taking care of these patient."

[Group 1, certified nurse assistant]

HCWs working in the community (e.g., home care settings), in particular, stopped visiting their patients, which led to drastic changes in their responsibilities by switching to telework. This change included teaching their patients to use technology and to share resources remotely.

"I've also seen that many people have felt like a little bit lost because with all of this involving technology, trying to talk via the internet has been something difficult for the elderly, and communicating with doctors."

[Group 2, community health worker]

In that context, those bilingual HCWs were also faced with assisting non-English speakers to adapt to new technology tools. They provided help to non-English speakers, particularly Spanish speakers, to sign in and register using various technology tools including apps and online appointments for testing. They also pointed at the difficulties identifying issues that can be done only by in-person visits such as domestic violence cases or lead exposure as described by one HCW serving in a home care setting.

"In this city, for example, lead in the paint. We would . . . go in physically, into the house and we'd see if the paint might have lead. . ."

[Participant 5, community health worker, home care setting]

In addition to changes in work responsibilities, participants described the loss of personal connections with patients and patient's family members including the inability to provide physical comfort. A participant in a direct care setting provided the following description:

"And then some residents, they want to get close to you, hold your hand. Their family is not around. They want to talk to you. . . . You can't do it no more."

[Participant 7, clinic coordinator, nursing home]

Participants also commented on the potential long-term changes in personal contact due to the pandemic.

"Human beings cannot only be limited to visual or audio contact. . . Do you understand me? Physical contact of, hello, a hug, a handshake. I think those things will never be the same after COVID."

[Participant 5, community health worker, home setting]

## Fears and uncertainty caused by the pandemic among HCWs

Participants described the daily fear and uncertainty they faced at work related to the risk of COVID-19 for them and their families. A participant reflected:

"It strikes me how sometimes I get up in the morning. I say I don't know what I'm going to face now. . . I have my children. We live together as a family. Sometimes when I come back from work I tell them, you know what? Don't come close to me. Let me get undressed, take my shower. Then when I'm ready, I'll come out and then we can say hi to each other."

[Participant 7, clinic coordinator, nursing home]

Participants also shared stories about the risks of working in direct care settings:

"You take care of this today. The next day, by the time you get to work, they told you they have the COVID. They have to move them. Then you say oh, my God. . . .You don't protect yourself, but I took care of this person yesterday and then it's like we are too close to each other."

[Participant 7, clinic coordinator, nursing home]

Concerns about the risk of losing their jobs or a portion of their income were also discussed.

"There was the fear of saying, yes, I'm positive, because obviously, they would immediately get sent home from work. . . .And obviously, the falling behind in bills or the fact that you did not work one day, gets you behind with all your bills. For example, rent payment, the electric bill."

[Participant 5, community health worker, home care setting]

They discussed the impact of informing their employers about possible exposure.

". . .they were exposed to someone–not at work, but on the ride to get to work. So, the managers tell them, okay, you were exposed outside of work to someone who was positive, so

you have to stay home and we're not going to pay you for those two weeks. . . they said, no, well, next time, I'm not going to notify them. I'm going to work because if they continue leaving me at home, in quarantine, with no pay, what am I going to do?"

(Group 1, community health worker)

Participants also brought up the issue of stigma among coworkers after testing positive.

"It's a huge stigma; not from everyone, but some people will make you feel very dirty, very uncomfortable, because someone has tested positive."

[Participant 6, administrative assistant in a hospital setting]

They also shared their own fear of coworkers who have tested positive.

"Right now, two people just came back—three—to our area that were tested positive and were out for a while, and it really gives me the creeps. I don't know. I'm thinking once it's in you, it's in you, and it's going to get me. So I try to stay as far as I can from them."

[Participant 10, dietary aid, nursing home setting]

## Shifts in testing and vaccines attitudes as the pandemic evolved

The rapid nature of the pandemic displayed the progression of testing in relation to frequency of testing, type of test, and testing procedures.

Some participants were provided with frequent testing, while others were tested less frequently or were not required to take tests at all.

"So it's basically your choice [testing]. It's not mandatory. It's not required."

[Participant 4, administrative support staff, hospital setting]

Another participant mentioned how they assumed the burden of testing by themselves to keep their families safe:

". . .I'd also do it every two weeks on my own behalf because if something happens to me, I didn't want to affect [my children]."

[Group 1, community health worker]

Additionally, employment status, such as contractors, who typically are third party employees, made some participants not eligible for testing. Such policies created logistical barriers and challenges for HCWs, as a participant explained:

"[S]ometimes I do find it difficult trying to figure out, am I eligible for the free testing that they have, and because I'm not a true employee, should I go somewhere else."

[Participant 6, administrative assistant, hospital setting]

Participants, in both direct and indirect care settings, also discussed temporal changes in frequency of testing and concerns about type of testing as the pandemic unfolded.

"Ours [test] is twice weekly for the lab, and then they had this thing where we were getting rapid tests three times per week, but that has stopped. . . .My main concern (. . .) I don't know if anybody else see it the way I do, but when it comes to rapid testing, that person is just tested and let go back into their work area until 15 minutes later. I don't think that is safe because more than once they find people that are positive, and they've already gone back to the people that it's like—it's—that's why I'm so scared of it.

[Participant 10, dietary aid, nursing home setting]

They also discussed their reasons for testing and changes in type of testing.

"I have to keep my parents safe. . . I was going to say one way of doing that is with testing. The hospital does not test us for COVID. Initially they did. They tested us for COVID. Then they tested us for the antibodies. We had nasal swab, then saliva, and then blood was drawn. And that was done all at once."

[Participant 9, physical therapy department staff, hospital setting]

Perspectives on vaccine skepticism and decisions around vaccination also evolved over time. Initial concerns about vaccines ranged from questions on secondary effects, trials data, and experiences of failed public health interventions in minority populations.

"Well, I just wanted to see the type of side effects, if there were any, other than just the mild temperature."

[Participant 8, outpatient care staff, hospital setting]

Participants discussed the evolution of their opinions about vaccines including how they initially were opposed to vaccination, but later changed their minds.

"Initially, it was a hard no. . . In the very, very beginning, I decided to let my coworkers go first and see what happened with them, and then I would do it. Wednesday evening, I finally logged on to make an appointment. . . .

[Participant 8, outpatient care staff, hospital setting]

They discussed reasons for changing their opinion about vaccines including learning about other coworkers taking the vaccine and acquiring vaccine data from reliable sources. They also shared how they investigated vaccine data themselves.

"I had questions. And so, part of my questions had to do with testing, believe it or not. . . Well, I'm not afraid to get it. It's not that. I have questions and I think it's fair that. . .I want my questions answered. So I actually couldn't get answers from anyone else, so that's when I wrote the NIH . . . I felt amazed that they answered me, and so quickly, and very specifically. And then I felt more comfortable about getting the vaccine.

[Participant 9, physical therapy department staff, hospital setting]

Some participants were vocal in their struggle about making decisions on the vaccine. They recognized their struggle between failed past public health strategies in minority populations and their current knowledge about science and prevention.

"Why are they offering it to Newark first? Is it because of the minorities? So they want to experiment on us, right? But then the intelligent part of me says that we should be blessed, and at the same time, if this is a chance to make it go away, then I have to do what I have to do, not for myself, but for my family."

[Participant 4, receptionist, hospital setting]

Finally, participants voiced their concern about vaccine mandates and the implication for their current employment.

"Either you are doing it or you are not getting your job. So I don't think they should—they kind of put us in a place whereby you have to choose between your job and the vaccine."

[Group 2, certified nurse assistant]

## Discussion

This study illustrated the multi-layered impact of the COVID-19 pandemic on Black and Latinx HCWs in ancillary and support roles. These workers received far less attention and recognition than doctors and nurses in the frontlines of the COVID-19 pandemic, yet they worked in similar high-risk settings and lived in the communities that suffered disproportionately. As such, their perspectives offer a unique lens on the pandemic including mitigation strategies like testing and vaccines that can inform policy.

HCWs reported a variety of changes in job responsibilities and personal connections. They also described the fear of contracting COVID-19 themselves and transmitting to their family members. These alterations led to continuous distress across HCWs in different roles. Worldwide, HCWs have described increased workloads, new tasks, and disruption on HCWs' ability to deliver on their usual work responsibilities [15, 20]. A study of palliative care workers including 41 countries, reported how the reorganization of work resulted in time-consuming tasks and accessibility barriers for those workers conducting home visits [15]. Home HCWs in New York City described their situation as a tough tradeoff between their own health and finances [14]. Our study showed that these alterations occurred across multiple settings including hospital, homes, nursing homes and community. Before the pandemic, all HCWs were known to be at high risk for anxiety, depression, burnout, insomnia, moral distress, and post-traumatic stress disorder [21, 22]. Our findings illustrate the critical need for health systems to provide targeted programs that support this marginalized workforce to mitigate the devastating impact of the pandemic on this group, promote healing, reduce burnout, and enhance retention.

In 2020, more people were employed in health care support roles than in all health care practitioners and technician jobs (doctors, nurses, emergency medical technicians [EMT], laboratory technicians) [7]. These HCWs provide frontline essential care, yet they are poorly compensated. In 2018, HCW, who are mostly women (>80%) and disproportionately Black and Latinx, made a median of $13.38 per hour with home health and personal care workers making only $11.52 per hour [23]. Furthermore, nearly 20% live in poverty and more than 40% rely on some public assistance [23]. This pandemic has exposed the stress endured by an essential workforce that plays a critical role in healthcare. As such, healthcare systems need to dedicate resources to improve the work conditions to this marginalized workforce. Working conditions can include pay, space, physical conditions and mental demands, health, safety and wellbeing among others [24]. Overall work conditions must be largely improved to be able to ensure their wellbeing and role retention to manage the next public health emergency.

Vaccine hesitancy has been widely discussed since early in the pandemic [25, 26]. Various surveys have reported shifts in vaccine hesitancy and enthusiasm among the U.S. population. For example, individuals taking the "wait and see approach" decreased by 8% between December 2020 and January 2021 [27]. Among HCW, our data suggest that access to information about the vaccine, candid answers to their questions, and seeing coworkers vaccinated all influenced their decisions on moving towards vaccination. However, some remained hesitant about vaccines citing distrust in the government and institutions based on past failed interventions in Black and Latinx populations. As of October 2021, 4 in 10 of all HCWs have not been vaccinated [28]. In New Jersey, the census estimates that 14% and 21% of the general population in 2019 was Black and Latinx, respectively. However, as of November 1st 2021, in New Jersey, Black and Latinx individuals represent only 8% and 16% of those fully vaccinated [29]. Our research suggests that transparent dialogue directly addressing questions and concerns about the COVID-19 vaccine by trusted entities or individuals may help to increase the number of vaccinated individuals within the HCW workforce. Participants also shared how seeing peers vaccinated influenced their decision to seek vaccination. The role of HCWs serving as ambassadors to provide accurate COVID-19 information and improve the number of vaccinated individuals among their coworkers and communities warrants additional study. This role may be particularly relevant in light of concerns voiced by HCWs in relation to vaccine mandates. It is unclear the extent to which states and/or employers might implement COVID-19 vaccine mandates, and whether or not mandates are the most effective means to achieve higher vaccination rates is unknown.

Our work has limitations. First, we sampled Black and Latinx participants from largely urbanized counties in one state; thus, our results may not transfer to other racial/ethnic groups or rural settings. We were, however, able to capture a unique, diverse population and HCWs from a variety of work settings including hospital, home, and community settings. Second, while we were able to capture temporal changes in testing and vaccine hesitation from November to February, given the rapid evolution of the pandemic it is likely that perspectives have continued to change. Nevertheless, this study design did enable us to capture participants' experiences at a critical juncture during the pandemic when the first rollout of vaccines was occurring.

## Conclusion

Our study illustrates the profound impact of the COVID-19 pandemic in Black and Latinx HCWs in ancillary and support roles. Disruption in their daily responsibilities and roles, abrupt structural changes, and fear of contracting COVID-19 caused continuous distress. Efforts to further examine the role of HCWs as ambassadors to improve the number of vaccinated individuals among their coworkers warrants additional research. This marginalized workforce has been an integral part of the fight against COVID-19; however, these workers remain underpaid and under recognized. Health systems must work to improve work conditions for this marginalized group to ensure their wellbeing and support their critical role in our communities during this pandemic and future public health emergencies.

## Supporting information

**S1 Data.**
(DOCX)

## Acknowledgments

We thank each of the participants without whom this study would not be possible. The authors are grateful to our community partners and Rutgers colleagues who comprise the NJ HEROES

TOO team including the following individuals and the organizations they represent: ASPIRA Inc., of New Jersey, Robyn D'Oria, Judith Francis, and Laura Taylor (Central Jersey Family Health Consortium), Dr. Pamela B. Jones (Communities in Cooperation), Barbara Booker, Rita Butts, Wilda Hobbs, and Tania Williams-Cajuste (East Orange Senior Volunteer Corporation), Kimberly M. Birdsall, MPH (Health Coalition of Passaic County), Mary R. Dawkins, Alfuguan Hardy, Amber Jennings, and Mayor Dahlia O. Vertreese (Hillside Senior Recreation Center), Harry Garcia, Paul Messer, Jr., and Ralph Stowe (Jazz4PCA), Megan Carduner, Rosela Roman, and Rosmery Suarez (Mobile Family Success Center), Toni Hendrix, Bruce Morgan Sr., and Deborah Morgan (New Brunswick Area Branch NAACP), Manuel J. Castañeda, Jaymie Santiago, and Staff (New Brunswick Tomorrow), Pastor Joe A. Carter, Francis J. Dixon, and Kelvin Roberson (New Hope Baptist Church), Roberto Muñiz (Parker Health Group, Inc.), Mayra Ramirez and Mariekarl Vilceus-Talty (Partnership for Maternal and Child Health of Northern New Jersey), Aitza Elhuni, José Carlos Montes, Carmelo Cintrón Vivas and programs team (PRAB), Dr. Beverly Lynn and Kendra Orta (Programs for Parents), Mariam Merced, MA (Robert Wood Johnson University Hospital), Uzo Achebe, Maria Ortiz, Tress Parker, and Dorothy Reed (Sister2Sister), Beverly Canady, Latisha Miller, Lou Schwarcz, and Leanna Waller (The Bridge, Inc.), James Horne and Juanita Vargas (United Way of Greater Union County), Dr. Chris Pernell, MD, MPH, FACPM (University Hospital-Newark), Donna L. Alexander and Kathy Waters (Urban League of Union County, Inc.), and Tami M. Videon (VNA Health Group).

## Author Contributions

**Conceptualization:** Zorimar Rivera-Núñez, Manuel E. Jimenez, Maria B. Pellerano, Jeanne M. Ferrante, Emily S. Barrett, Martin J. Blaser, Reynold A. Panettieri, Jr., Shawna V. Hudson.

**Formal analysis:** Zorimar Rivera-Núñez, Manuel E. Jimenez, Benjamin F. Crabtree, Brittany Sullivan, Shawna V. Hudson.

**Funding acquisition:** Manuel E. Jimenez, Diane Hill, Emily S. Barrett, Martin J. Blaser, Reynold A. Panettieri, Jr., Shawna V. Hudson.

**Investigation:** Shawna V. Hudson.

**Methodology:** Benjamin F. Crabtree, Maria B. Pellerano, Jeanne M. Ferrante, Martin J. Blaser, Reynold A. Panettieri, Jr., Shawna V. Hudson.

**Project administration:** Donita Devance, Myneka Macenat, Daniel Lima, Marsha Gordon, Brittany Sullivan.

**Supervision:** Zorimar Rivera-Núñez, Manuel E. Jimenez, Benjamin F. Crabtree, Shawna V. Hudson.

**Writing – original draft:** Zorimar Rivera-Núñez, Manuel E. Jimenez.

**Writing – review & editing:** Zorimar Rivera-Núñez, Manuel E. Jimenez, Benjamin F. Crabtree, Diane Hill, Maria B. Pellerano, Donita Devance, Myneka Macenat, Daniel Lima, Marsha Gordon, Brittany Sullivan, Robert J. Rosati, Jeanne M. Ferrante, Emily S. Barrett, Martin J. Blaser, Reynold A. Panettieri, Jr., Shawna V. Hudson.

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
