## [Decision Letter · Decision Letter 0]

28 Oct 2021

PONE-D-21-19556Experiences of Black and Latinx health care workers in support roles during the COVID-19 pandemic: A qualitative studyPLOS ONE

Dear Dr. Rivera-Nunez,

Thank you for submitting your manuscript to PLOS ONE. After careful consideration, we feel that it has merit but does not fully meet PLOS ONE’s publication criteria as it currently stands. Therefore, we invite you to submit a revised version of the manuscript that addresses the points raised during the review process.Please submit your revised manuscript within 60 days of receipt of this email. If you will need more time than this to complete your revisions, please reply to this message or contact the journal office at plosone@plos.org. Please include the following items when submitting your revised manuscript:A rebuttal letter that responds to each point raised by the academic editor and reviewer(s). You should upload this letter as a separate file labeled 'Response to Reviewers'.A marked-up copy of your manuscript that highlights changes made to the original version. You should upload this as a separate file labeled 'Revised Manuscript with Track Changes'.An unmarked version of your revised paper without tracked changes. You should upload this as a separate file labeled 'Manuscript'.

We look forward to receiving your revised manuscript.

Kind regards,

Marlene Camacho-Rivera, ScD, MPH

Academic Editor

PLOS ONE

Journal Requirements:

a) Did participants provide their written or verbal informed consent to participate in this study?

3. Thank you for stating the following in the Competing Interests/Financial Disclosure* (delete as necessary) section:

“I have read the journal's policy and the authors of this manuscript have the following competing interests: Research reported in this publication was supported by grant UL1TR003017 from the National Center for Advancing Translational Sciences at the National Institutes of Health (NIH). NIH had no role in the design and conduct of the study; collection, management, analysis and interpretation of the data; preparation, review or approval of the manuscript; and decision to submit the manuscript for publication. Ms. Pellerano reports grant funding from Johnson & Johnson Corporate Foundation, personal fees from the Patient Centered Outcome Research Institute for grant reviewing, and personal fees from the University of Massachusetts, Lowell for participation on a grant advisory committee. The authors have no other financial interests to disclose.”

We note that one or more of the authors are employed by a commercial company: NIH, Johnson & Johnson Corporate Foundation

5. PLOS requires an ORCID iD for the corresponding author in Editorial Manager on papers submitted after December 6th, 2016. Please ensure that you have an ORCID iD and that it is validated in Editorial Manager. To do this, go to ‘Update my Information’ (in the upper left-hand corner of the main menu), and click on the Fetch/Validate link next to the ORCID field. This will take you to the ORCID site and allow you to create a new iD or authenticate a pre-existing iD in Editorial Manager. Please see the following video for instructions on linking an ORCID iD to your Editorial Manager account: https://www.youtube.com/watch?v=_xcclfuvtxQ"

“This work was supported by the National Center for Advancing Translational Sciences UL1TR003017 at the National Institutes of Health.”

“The funders had no role in study design, data collection and analysis, decision to publish, or preparation of the manuscript”

Additionally, because some of your funding information pertains to [commercial funding//patents], we ask you to provide an updated Competing Interests statement, declaring all sources of commercial funding.

In your Competing Interests statement, please confirm that your commercial funding does not alter your adherence to PLOS ONE Editorial policies and criteria by including the following statement: "This does not alter our adherence to PLOS ONE policies on sharing data and materials.” as detailed online in our guide for authors  http://journals.plos.org/plosone/s/competing-interests.  If this statement is not true and your adherence to PLOS policies on sharing data and materials is altered, please explain how.

Please include the updated Competing Interests Statement and Funding Statement in your cover letter. We will change the online submission form on your behalf.

7. We note that the grant information you provided in the ‘Funding Information’ and ‘Financial Disclosure’ sections do not match.

Reviewers' comments:

Reviewer's Responses to Questions

**Comments to the Author**

1. Is the manuscript technically sound, and do the data support the conclusions?

Reviewer #1: Yes

2. Has the statistical analysis been performed appropriately and rigorously? 

Reviewer #1: N/A

3. Have the authors made all data underlying the findings in their manuscript fully available?

Reviewer #1: Yes

4. Is the manuscript presented in an intelligible fashion and written in standard English?

Reviewer #1: Yes

5. Review Comments to the Author

Reviewer #1: It has been well documented that individuals from minority populations have been disproportionately impacted by the COVID-19 pandemic. One such example is provided within the context of poor health outcomes in minority populations. Among Black and Latinx individuals the risk of hospitalization and death from COVID-19 are 2-3 times higher when compared to their white counterparts. The intersectionality of factors such as structural and institutional racism, their higher risk factors and the reduced ability to avoid infection as essential workers make Black and Latinx health care workers (HCWs) a population that is both vulnerable and high-risk.

This manuscript presents a qualitative study that examined the experiences of Black and Latinx HCWs to understand how the pandemic impacted their lives in the following areas: profession, job responsibilities and relationship with their communities. These findings are particularly relevant given that as of September 2021, there exist very few (if any) peer-reviewed published literature that focuses on the unique dual perspective of participants that are both members of a high-risk essential workforce and are also members of marginalized communities that suffered disproportionately during the pandemic.

Participants of this qualitative study were employed as ancillary and support staff at four health care facilities (various inpatient and outpatient health care settings) in New Jersey. It is notable that each of the health care facilities was geographically located in four counties, each with large Black/Latinx populations and high levels of COVID-19. Semi-structured group and individual interviews were conducted virtually using a Zoom secure platform. Proof of the iterative process in developing the semi-structured interview guide was demonstrated through the study’s incorporation of literature review, prior experience and debriefing meetings that were conducted after initial interviews. The interview data was analyzed using an editing approach as it was collected. Three key themes were identified: (1) Profound impact of the pandemic on job responsibilities, work settings and personal connections, (2) Fear and uncertainties caused by the pandemic; and, (3) Shifts in testing frequency and vaccine attitudes as the pandemic evolved. These findings revealed that perspectives expressed by Black and Latinx HCWs were remarkably consistent and informed the authors’ conclusion of reaching thematic saturation.

I found the study to be a well-written and much needed investigation to provide specific insight into the perspectives and experiences of this unique population who are both members of a high-risk essential workforce and are also members of marginalized communities that have been most heavily impacted by COVID-19. I found no major issues with this manuscript. Additionally, it is important to note the context/timing of when this study was conducted is significant (Nov 2020-Feb 2021.) It was conducted during a pivotal moment in the pandemic, which coincided with the first rollout of COVID-19 vaccines. The study’s findings successfully captured temporal changes in HCW experiences at the workplace, including testing and evolving perspectives regarding vaccine hesitation. The participants’ responses provided unique insight into many of the barriers that Black and Latinx HCWs, patients and community members alike, encounter with regard to: essential worker workplace challenges, the availability and accessibility of COVID-19 testing and vaccines. The authors’ address various limitations of the study, specifically the limitations of the sample population in that they “purposively sampled only Black and Latinx HCWs…” from largely urbanized counties in one state. Another minor, yet potential limitation is the relatively small (n=17) and homogeneous sample population. However, as stated, this was done intentionally by the authors’ and does indeed capture highly specific data on a very distinct population of “dual perspective” all-female Black and Latinx HCW participants. Another minor issue is in Lines 139-140, referencing Table 1. Line 140 states that 91% percent of the sample population was female. However, Table 1 reflects that of n=17, 100% is female. Line 140 also states that 82% of HCWs were Black and 18% were Latinx. Yet, Table 1 reflects that 47.1% were Black and 52.9% were Latinx. Again, these are very minor issues and can be easily revised by the authors. I very much enjoyed reading this manuscript and wish to congratulate the authors on this well-written and thought provoking work.

6. PLOS authors have the option to publish the peer review history of their article (what does this mean?). If published, this will include your full peer review and any attached files.

Reviewer #1: **Yes: **Sara Mizany, MS

---

## [Author Response · Author response to Decision Letter 0]

8 Nov 2021

Response to Reviewers

Reviewer #1

I found the study to be a well-written and much needed investigation to provide specific insight into the perspectives and experiences of this unique population who are both members of a high-risk essential workforce and are also members of marginalized communities that have been most heavily impacted by COVID-19. I found no major issues with this manuscript. Additionally, it is important to note the context/timing of when this study was conducted is significant (Nov 2020-Feb 2021.) It was conducted during a pivotal moment in the pandemic, which coincided with the first rollout of COVID-19 vaccines. The study’s findings successfully captured temporal changes in HCW experiences at the workplace, including testing and evolving perspectives regarding vaccine hesitation. The participants’ responses provided unique insight into many of the barriers that Black and Latinx HCWs, patients and community members alike, encounter with regard to: essential worker workplace challenges, the availability and accessibility of COVID-19 testing and vaccines. The authors’ address various limitations of the study, specifically the limitations of the sample population in that they “purposively sampled only Black and Latinx HCWs…” from largely urbanized counties in one state. Another minor, yet potential limitation is the relatively small (n=17) and homogeneous sample population. However, as stated, this was done intentionally by the authors’ and does indeed capture highly specific data on a very distinct population of “dual perspective” all-female Black and Latinx HCW participants. 

1. Another minor issue is in Lines 139-140, referencing Table 1. Line 140 states that 91% percent of the sample population was female. However, Table 1 reflects that of n=17, 100% is female. 

Thank you for identifying this inconsistency in our manuscript. We appreciate the opportunity of correcting it. We have corrected the % and updated the manuscript as follow: Table 1 summarizes demographics and job descriptions of our sample. The median age was 48 years (range 25-58); 47% were Black, 53% Latinx, and 100% female. Lines 140-141

2. Line 140 also states that 82% of HCWs were Black and 18% were Latinx. Yet, Table 1 reflects that 47.1% were Black and 52.9% were Latinx. Again, these are very minor issues and can be easily revised by the authors. I very much enjoyed reading this manuscript and wish to congratulate the authors on this well-written and thought provoking work.

Thank you again for identifying this inconsistency in our manuscript. We have corrected the % and updated the manuscript as follow: Table 1 summarizes demographics and job descriptions of our sample. The median age was 48 years (range 25-58); 47% were Black, 53% Latinx, and 100% female. Lines 140-141.

---

## [Decision Letter · Decision Letter 1]

31 Dec 2021

Experiences of Black and Latinx health care workers in support roles during the COVID-19 pandemic: A qualitative study

PONE-D-21-19556R1

Dear Dr. Rivera-Nunez,

We’re pleased to inform you that your manuscript has been judged scientifically suitable for publication and will be formally accepted for publication once it meets all outstanding technical requirements.

Kind regards,

Marlene Camacho-Rivera, ScD, MPH

Academic Editor

PLOS ONE

Additional Editor Comments (optional):

Reviewers' comments:

Reviewer's Responses to Questions

**Comments to the Author**

1. If the authors have adequately addressed your comments raised in a previous round of review and you feel that this manuscript is now acceptable for publication, you may indicate that here to bypass the “Comments to the Author” section, enter your conflict of interest statement in the “Confidential to Editor” section, and submit your "Accept" recommendation.

Reviewer #1: All comments have been addressed

2. Is the manuscript technically sound, and do the data support the conclusions?

Reviewer #1: Yes

3. Has the statistical analysis been performed appropriately and rigorously? 

Reviewer #1: N/A

4. Have the authors made all data underlying the findings in their manuscript fully available?

Reviewer #1: Yes

5. Is the manuscript presented in an intelligible fashion and written in standard English?

Reviewer #1: Yes

6. Review Comments to the Author

Reviewer #1: (No Response)

7. PLOS authors have the option to publish the peer review history of their article (what does this mean?). If published, this will include your full peer review and any attached files.

Reviewer #1: **Yes: **Sara Mizany

---

## [Editor Report · Acceptance letter]

7 Jan 2022

PONE-D-21-19556R1 

Experiences of Black and Latinx health care workers in support roles during the COVID-19 pandemic: A qualitative study 

Dear Dr. Rivera-Núñez:

I'm pleased to inform you that your manuscript has been deemed suitable for publication in PLOS ONE. Congratulations! Your manuscript is now with our production department. 

Kind regards, 

on behalf of

Dr. Marlene Camacho-Rivera 

Academic Editor

PLOS ONE